# Structural insights into selective interaction between type IIa receptor protein tyrosine phosphatases and Liprin-α

Maiko Wakita[1,2,3,9], Atsushi Yamagata[1,2,3,4,7,9], Tomoko Shiroshima[1,2,4], Hironori Izumi[5], Asami Maeda[1,2,4], Mizuki Sendo[5], Ayako Imai[5], Keiko Kubota[2], Sakurako Goto-Ito[1,2,4], Yusuke Sato[1,2,3,4,8], Hisashi Mori [5], Tomoyuki Yoshida[5,6]* & Shuya Fukai [1,2,3,4]*

Synapse formation is induced by transsynaptic interaction of neuronal cell-adhesion molecules termed synaptic organizers. Type IIa receptor protein tyrosine phosphatases (IIa RPTPs) function as presynaptic organizers. The cytoplasmic domain of IIa RPTPs consists of two phosphatase domains, and the membrane-distal one (D2) is essential for synapse formation. Liprin-α, which is an active zone protein critical for synapse formation, interacts with D2 via its C-terminal domain composed of three tandem sterile alpha motifs (tSAM). Structural mechanisms of this critical interaction for synapse formation remain elusive. Here, we report the crystal structure of the complex between mouse PTPδ D2 and Liprin-α3 tSAM at 1.91 Å resolution. PTPδ D2 interacts with the N-terminal helix and the first and second SAMs (SAM1 and SAM2, respectively) of Liprin-α3. Structure-based mutational analyses in vitro and in cellulo demonstrate that the interactions with Liprin-α SAM1 and SAM2 are essential for the binding and synaptogenic activity.

[1] Institute for Quantitative Biosciences, The University of Tokyo, Tokyo 113-0032, Japan. [2] Synchrotron Radiation Research Organization, The University of Tokyo, Tokyo 113-0032, Japan. [3] Department of Computational Biology and Medical Sciences, Graduate School of Frontier Sciences, The University of Tokyo, Chiba 277-8561, Japan. [4] CREST, JST, Saitama 332-0012, Japan. [5] Department of Molecular Neuroscience, Graduate School of Medicine and Pharmaceutical Sciences, University of Toyama, Toyama 930-0194, Japan. [6] PRESTO, JST, Saitama 332-0012, Japan. [7] Present address: RIKEN Center for Biosystems Dynamics Research, Kanagawa 230-0045, Japan. [8] Present address: Center for Research on Green Sustainable Chemistry, Tottori University, Tottori 680-8582, Japan. [9] These authors contributed equally: Maiko Wakita, Atsushi Yamagata *email: toyoshid@med.u-toyama.ac.jp; fukai@iam.u-tokyo.ac.jp

Synaptic organizers are a class of neuronal adhesion or secretory proteins capable of inducing synaptic differentiation, and play important roles in formation and maturation of neuronal synapses[1,2]. Type IIa receptor protein tyrosine phosphatases (IIa RPTPs) function as presynaptic organizers. Dysfunctions of IIa RPTPs and their postsynaptic partners are associated with neurodevelopmental disorders, such as autism spectrum disorders (ASD), intellectual disability, or schizophrenia[1,3,4]. The vertebrate IIa RPTP family consists of three members, PTPδ, PTPσ, and LAR, which share the same domain architecture with a large N-terminal extracellular domain, a single transmembrane segment, and a cytoplasmic domain (Fig. 1a)[3,4]. The N-terminal extracellular domain comprises three immunoglobulin-like (Ig) domains and four or eight fibronectin type III (FN) domains[5]. The extracellular domain mediates a heterophilic, transsynaptic interaction with various postsynaptic organizers such as Netrin-G ligand 3 (NGL-3)[6,7], Tropomyosin kinase C (TrkC)[8], Interleukin-1 receptor accessory protein-like 1 (IL1RAPL1)[9], Interleukin-1 receptor accessory protein (IL-1RAcP)[10], Slit- and Trk-like family protein (Slitrk) 1–Slitrk6[11,12], synaptic adhesion-like molecule (SALM) 3[13], and SALM5[14]. All of these postsynaptic organizers except NGL-3 interact with the Ig domains of IIa RPTPs[15–20]. Their selective interactions are controlled by two mini-exon-derived peptides (meA and meB peptides), which are located within the Ig2 domain and in the junction between the Ig2 and Ig3 domains, respectively.

The cytoplasmic domain of IIa RPTPs comprises two tandem phosphatase domains[3,4]. The membrane-proximal phosphatase domain (D1) is catalytically active, whereas the membrane-distal one (D2) is inactive. The D2 domain of IIa RPTPs directly interacts with several synaptic proteins including Liprin-α[21,22], Caskin[23], and Trio[24]. In heterologous synaptogenic co-culture assays using short hairpin RNA (shRNA)-mediated PTPσ knockdown neurons and HEK293T cells expressing TrkC or Slitrk1, re-expression of phosphatase-dead PTPσ mutants could partially recover the synaptogenic activity in PTPσ-knockdown neurons but that of a D2-lacking mutant could not[25]. The intracellular interactions of PTPσ with synaptic proteins via the D2 domain likely play more critical roles in synaptogenesis than the phosphatase activity catalyzed by the D1 domain.

Liprin-α is an active zone protein and the first protein identified as an intracellular binding partner of IIa RPTPs[21]. Liprin-α belongs to the Liprin family, which is highly conserved from invertebrates to mammals[22,26]. The vertebrate Liprin family is classified into three groups: Liprin-α, Liprin-β, and KazrinE[22,27]. Mammals possess four Liprin-α isoforms (Liprin-α1–Liprin-α4), two Liprin-β isoforms (Liprin-β1 and -β2), and KazrinE. On the other hand, invertebrates possess a single set of Liprin-α, -β, and -γ[22,28]. Liprin-γ is the closest homolog of KazrinE. Among them, Liprin-α and -γ can reportedly bind to the D2 domain of IIa RPTPs. The existence of only a single *liprin-α* gene in invertebrates makes it simpler to assess in vivo function of Liprin-α in the nervous system. A loss of function in *syd-2* gene (Liprin-α homolog in *Caenohabiditis elegans*) alters the size and shape of presynaptic termini[29–32]. Deletion of the IIa RPTP homolog, *ptp-3*, resulted in mislocalization of SYD-2. In *Drosophila*, Dliprin-α, a homolog of Liprin-α, is required for normal synaptic morphology at neuromuscular junction and for photoreceptor target selection[28,33,34].

Although all four isoforms of Liprin-α are expressed in the brain with differential distribution and increase the complexity of their neuronal functions, Liprin-α2 and -α3 are predominantly and specifically expressed in the brain[22,35,36]. Knockdown of Liprin-α2 by shRNA alters synaptic vesicle pool size and presynaptic ultrastructure. Liprin-α2 regulates the turnover of

the active zone proteins, calcium/calmodulin-dependent serine kinase (CASK) and RIM1/2, to facilitate synaptic transmission[37]. Superresolution microscopy revealed substantial difference in the

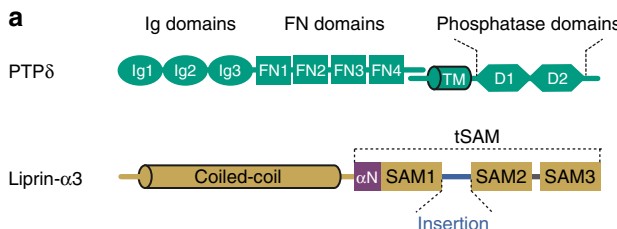

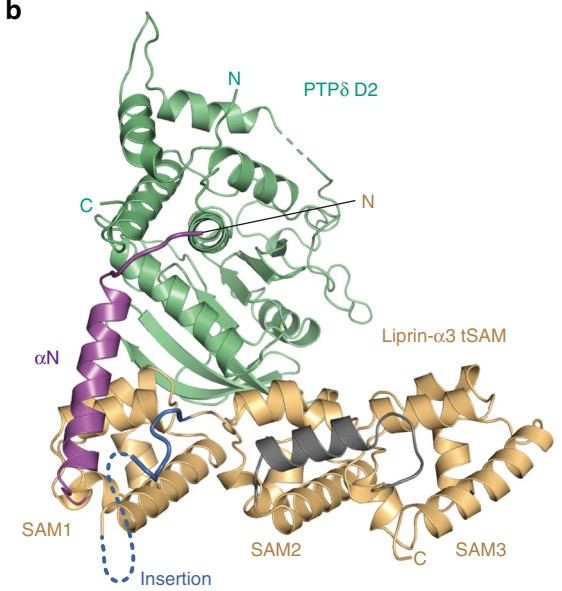

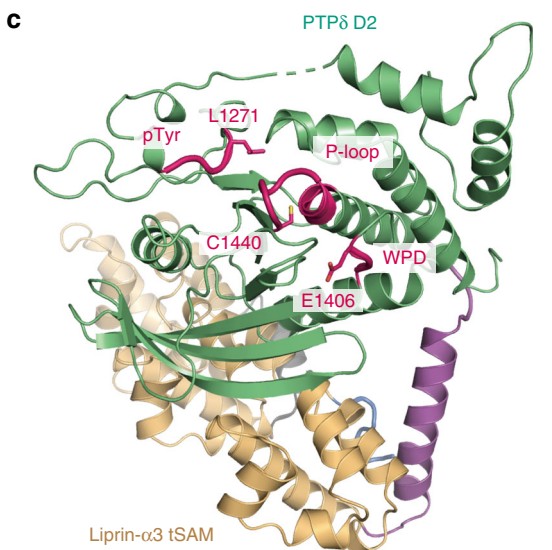

**Fig. 1 Structure of the complex between PTPδ D2 and Liprin-α3 tSAM.** **a** Domain organizations of PTPδ and Liprin-α3 (Ig, Immunoglobulin-like; FN, Fibronectin type III; TM, transmembrane; SAM, sterile alpha motif; tSAM, tandem SAM). **b** Overall structure of the complex between PTPδ D2 (green) and Liprin-α3 tSAM (αN, purple; SAM1–SAM3, light brown; the insertion between SAM1 and SAM2, dark blue; the linker helix between SAM2 and SAM3, dark gray). Disordered regions are shown as dotted lines. **c** Pseudocatalytic site of PTPδ D2. The coloring scheme is the same as that in **b**, except that the pTyr, P-loop, and WPD regions are colored in magenta.

localization of Liprin-α2 and -α3 inside the presynapse[38]. Liprin-α3 is substantially colocalized with the active zone proteins, whereas Liprin-α2 is localized to more internal region of the nerve terminals. Liprin-α3-knockout mice generated by CRISPR/Cas9 gene editing show impaired exocytosis of synaptic vesicles, although they survive and develop mostly normally, possibly due to a complementary function of Liprin-α2[38]. In fact, the depletion of Liprin-α3 causes the translocation of Liprin-α2 to the acive zone, though their functions may not be completely overlapped[37,38].

The Liprin family shares similar domain architecture comprising an N-terminal coiled-coil domain and three tandem sterile alpha motifs (SAM1–SAM3; hereafter referred to as tSAM; Fig. 1a). The N-terminal coiled-coil domain of Liprin-α mediates the binding to CAST/ELKS[39], GIT1[40], RIM[37], KIF1A[41], and mDia[42,43]. The tSAM domain of Liprin-α interacts with the D2 domain of IIa RPTPs[21,22] and CASK[44]. In addition, Liprin-α and -β form a heterodimer through the interaction between Liprin-α SAM1 and Liprin-β SAM3[22,45]. A wide variety of binding partners for Liprin-α imply its broad synaptic functions.

The complex structure between the tSAM domain of Liprin-α2 and CASK revealed that the insertion loop between SAM1 and SAM2 in Liprin-α2 interacts with CASK[45]. Notably, this insertion loop is only found in the vertebrate Liprin family. In the complex structure between the tSAM domains of Liprin-α2 and -β1, their heterodimeric assembly was mediated by their SAM1 and SAM3 motifs, respectively[45]. On the other hand, no structural information of the IIa RPTP–Liprin-α complex is available, although their interaction is likely to be the first important intracellular process for inducing presynaptic differentiation. Here, we present the crystal structure of the complex between PTPδ D2 and the tSAM domain of Liprin-α3. The D2 domain of PTPδ interacts with the N-terminal helix (αN), SAM1, and SAM2 in the tSAM domain of Liprin-α3. The structure-based mutational analyses in vitro and in cellulo demonstrate that the interaction with the SAM1 and SAM2 domains of Liprin-α3 is essential for the binding and synaptogenic activity.

## Results

**Overall structure of the PTPδ D2–Liprin-α3 tSAM complex.** To reveal the structural basis of the interaction between IIa RPTPs and Liprin-α, we sought to determine the crystal structure of the complex between the intracellular domain of IIa RPTPs and the tSAM domain of Liprin-α. The D1–D2 and D2 domains of mouse PTPδ and PTPσ and the tSAM domains of mouse Liprin-α1, -α2, -α3, and -α4 were purified and subjected to co-crystallization screening. Their expression constructs were designed, based on the structures of Liprin-α2 tSAM[45] and the phosphatase domains of IIa RPTPs[46]. Consequently, the combination of PTPδ D2 and Liprin-α3 tSAM yielded diffraction-quality crystals. The crystal structure of the PTPδ D2–Liprin-α3 tSAM complex was determined by the molecular replacement method using PTPσ D2 (PDB 2FH7 [https://doi.org/10.2210/pdb2FH7/pdb])[46] and Liprin-α2 tSAM (PDB 3TAD [https://doi.org/10.2210/pdb3TAD/pdb]; the CASK-interacting loop between SAM1 and SAM2 was trimmed)[45] as the search models. The asymmetric unit contains one complex, and no obvious molecular contact suggestive of higher-order assembly was found in the crystal (Supplementary Fig. 1). The final model was refined at 1.91 Å resolution with $R_{free}$ of 0.208 (Table 1 and Fig. 1a, b). The residue numbering of PTPδ in this paper is based on a mouse PTPδ isoform (NM_011211.3 [https://www.ncbi.nlm.nih.gov/nuccore/NM_011211.3]), which contains four FN domains with the 9-residue meA and meB peptide insertions (A9B+).

| **Table 1 Data collection and refinement statistics.** | |
|---|---|
| | **PTPδ D2–Liprin-α3 tSAM (PDB 6KIP)** |
| **Data collection** | |
| Beamline | SPring-8 BL41XU |
| Space group | $P4_12_12$ |
| Cell dimensions | |
| $a, b, c$ (Å) | 98.0, 98.0, 140.0 |
| $\alpha, \beta, \gamma$ (°) | 90.0, 90.0, 90.0 |
| Resolution (Å) | 50–1.91 (1.94–1.91) |
| $R_{sym}$ | 0.105 (0.278) |
| $I/\sigma I$ | 22.1 (2.3) |
| Completeness (%) | 97.3 (91.2) |
| Redundancy | 10.2 (3.5) |
| **Refinement** | |
| Resolution (Å) | 1.91 |
| No. reflections | 52,074 (2,644) |
| $R_{work}/R_{free}$ | 0.162/0.208 |
| No. atoms | |
| Protein | 4,588 |
| Water | 428 |
| $B$ factors (Å$^2$) | |
| Protein | 35.2 |
| Water | 40.3 |
| R.m.s. deviations | |
| Bond lengths (Å) | 0.007 |
| Bond angles (°) | 1.020 |

Data were collected from a single crystal. Highest-resolution shell is in parentheses.

The D2 domain of PTPδ adopts an α/β structure, which closely resembles the reported D1 and D2 structures of IIa RPTPs (Cα rmsds of 0.59–1.23 Å over 244–287 residues; Supplementary Table 1)[46,47]. The pseudoactive site of PTPδ D2 retains the catalytic cysteine residue (Cys1440). However, the tyrosine residue in the phosphorylated Tyr recognition loop and the aspartate residue in the WPD motif, which are critical for the phosphatase activity[48,49], are replaced with Leu and Glu, respectively. These replacements synergistically reduce the phosphatase activity[47,50]. The interface between Liprin-α3 and PTPδ D2 is located on the opposite side of the pseudoactive site (Fig. 1c).

The tSAM domain of Liprin-α3 consists of three SAM domains, SAM1–SAM3. The additional helix is located at the N-terminal end of SAM1 (αN). SAM2 and SAM3 are connected by the short linker helix (Fig. 1a, b). The electron density of the CASK-interacting loop connecting SAM1 and SAM2 was invisible. The individual SAM structures and their relative configurations are similar between PTPδ D2-bound Liprin-α3 and CASK-bound Liprin-α2 (Cα rmsd of 1.39 Å over 258 residues; Supplementary Table 1)[45].

**Interactions between PTPδ D2 and Liprin-α3 tSAM.** The D2 domain of PTPδ interacts with αN, SAM1, and SAM2 of Liprin-α3 but not with SAM3 in the present crystal structure of the PTPδ D2–Liprin-α3 tSAM complex (Fig. 2a and Supplementary Table 2). At the interface with αN, Phe1503 and Tyr1506 of PTPδ hydrophobically interact with Leu808 of Liprin-α3, and Asp1504 of PTPδ forms hydrogen bonds with Arg816 of Liprin-α3 (Fig. 2b). At the interface with SAM1, Tyr1373, Leu1380, Phe1399, and Phe1430 of PTPδ form a hydrophobic pocket, which accommodates Trp856 of Liprin-α3 (Fig. 2c). At the interface with SAM2, Phe1430 of PTPδ hydrophobically interacts with Leu978 of Liprin-α3, whereas Arg1397 and Asp1433 of PTPδ form hydrogen bonds with Glu976 and Arg971 of Liprin-α3, respectively (Fig. 2d). Note that Phe1430 of PTPδ hydrophobically interacts with both SAM1 and SAM2 of Liprin-α3.

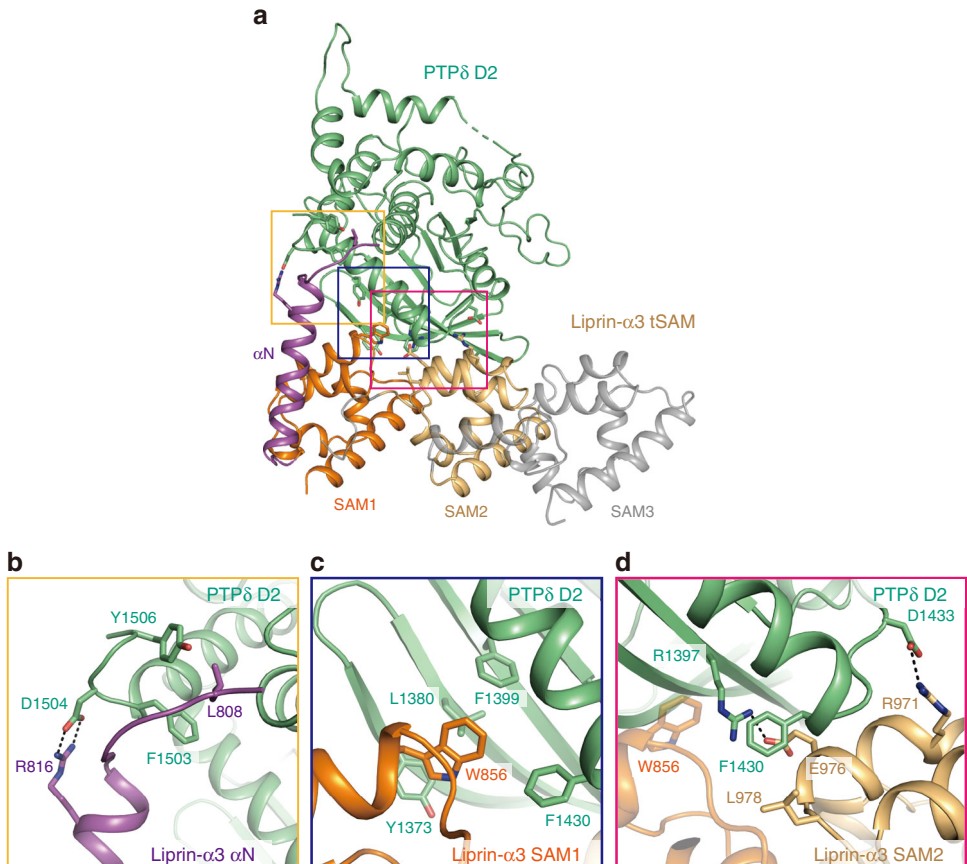

**Fig. 2 Binding interfaces between PTPδ D2 and Liprin-α3 tSAM. a** Overall view of the interaction between PTPδ D2 (green) and Liprin-α3 tSAM (αN, purple; SAM1, orange; SAM2, light brown; other regions, gray). Disordered regions are shown as dotted lines. The residues involved in the PTPδ D2–Liprin-α3 tSAM interaction are shown as sticks. Close-up views of the interactions of PTPδ D2 with αN (**b**), SAM1 (**c**), and SAM2 (**d**) of Liprin-α3 tSAM. The coloring scheme is the same as that in **a**. Hydrogen bonds are shown as dotted black lines.

**Table 2 Mutational analyses of the interaction between PTPδ D2 and Liprin-α3 tSAM by SPR experiments.**

| Mutation | $K_D$ (μM) |
| --- | --- |
| **Liprin-α3 tSAM** | |
| WT | 10.7 ± 3.3 |
| ΔN9 | 17.4 ± 0.15 |
| L808A | 19.7 ± 0.58 |
| R816A | 29.8 ± 1.3 |
| W856A | ND |
| E976A | >40 |
| L978A | ND |
| **PTPδ D2** | |
| F1503A | 8.3 ± 2.8 |
| D1504A | 18.0 ± 3.8 |
| Y1506A | 11.4 ± 2.3 |
| Y1373A | ND |
| L1380A | >40 |
| F1399A | 32.0 ± 6.1 |
| R1397A | ND |
| F1430A | ND |

Source data are provided as a Source Data file.
*WT* wild type, *ND* not detectable.

To assess the contribution of these intermolecular interactions to the binding affinity between PTPδ D2 and Liprin-α3 tSAM, site-directed mutants of PTPδ D2 or Liprin-α3 tSAM were analyzed by surface plasmon resonance (SPR)

spectroscopy (Table 2 and Supplementary Fig. 2). Unexpectedly, point mutations or 9-residue deletion (ΔN9) at the interface with αN showed little effects on the affinity between PTPδ D2 and Liprin-α3 tSAM. In contrast, mutations at the interfaces with SAM1 and SAM2 severely disrupted the binding. As for the mutations at the SAM1 interface, the W856A mutation of Liprin-α3 completely abolished the binding to PTPδ D2. Correspondingly, the Y1373A mutation of PTPδ almost abolished the binding, whereas the L1380A and F1399A mutations reduced the affinities >fourfold and threefold, respectively. As for the mutations at the SAM2 interface, the L978A and E976A mutations of Liprin-α3 almost abolished the binding and reduced the affinity >fourfold, respectively. The F1430A and R1397A mutations of PTPδ both impaired the binding. Taken together, the SAM1 and SAM2 interfaces are critical for the binding between PTPδ D2 and Liprin-α3 tSAM, whereas the αN interface contributes little to the binding. The key residues for the PTPδ D2–Liprin-α3 tSAM interaction are conserved in both IIa RPTPs and Liprin-α proteins from representative metazoa (Supplementary Figs. 3 and 4).

**Mechanisms for specificity between IIa RPTP D2 and Liprin-α.** The D2 domain of IIa RPTPs binds to Liprin-α proteins but not to Liprin-β proteins[21], despite their some sequence similarity (*e.g.*, 36% identity between mouse Liprin-α3 SAM1-SAM2 and Liprin-β1 SAM1-SAM2; the CASK-interacting loop of Liprin-α3 was excluded). Superposition of the structures of mouse Liprin-α3 and Liprin-β1 highlights that Trp856 and Leu978 of Liprin-α3, which

are critical for binding to PTPδ and conserved in the Liprin-α family, are replaced with Ser and Arg in Liprin-β1, respectively (Fig. 3a). Similar differences are found between the Liprin-α and Liprin-β families (Supplementary Fig. 5), explaining the specific binding of IIa RPTPs to Liprin-α.

A previous yeast two-hybrid assay showed that Liprin-α1 binds to the isolated LAR D2 but not to D1[21], although the D1 and D2 domains are topologically similar to each other[46,47]. To elucidate the mechanism for the specificity of Liprin-α to the D2 domain of IIa RPTPs, the D2 structure of PTPδ was compared with the D1 structure. Among the PTPδ D2 residues that were found to be critical for binding to Liprin-α in this study, Tyr1373, Leu1380, and Phe1430 of PTPδ D2 are not present in the corresponding positions of PTPδ D1 (Fig. 3b). Particularly, substantial structural difference is observed in the region around Phe1430 of PTPδ D2. At the interface between PTPδ D2 and Liprin-α3 SAM2, Phe1430 of PTPδ D2 is located on the edge of an α-helix. The corresponding position in the D1 domain of IIa RPTPs is occupied with the highly conserved proline residue, which shortens the α-helix so as to disable the interaction with Liprin-α.

The structural mechanism for the interaction between PTPδ D2 and Liprin-α3 tSAM, which we revealed in this study,

confirms the specific binding between the D2 domain of IIa RPTPs and the tSAM domain of Liprin-α proteins.

**Impact of PTPδ–Liprin-α interaction on presynapse formation.** Essential roles of Liprin-α in synapse formation have been demonstrated by knockout experiments of *C. elegans* and *D. melanogaster*, both of which have a single *liprin-α* gene. On the other hand, vertebrates have several Liprin-α isoforms (*e.g.*, four isoforms in mouse and human), and a complete shutdown of Liprin-α-mediated signaling likely requires a multiple gene knockout approach. In Liprin-α3-null mice, Liprin-α2 possibly compensates for the lack of Liprin-α3.

The present structural analysis of the interaction between PTPδ D2 and Liprin-α3 tSAM unveiled the critically important residues of IIa RPTPs for binding to Liprin-α proteins. The mutations of these residues may selectively shutdown the PTPδ–Liprin-α signaling axis for inducing presynaptic differentiation. To test this possibility, we examined the effects of the PTPδ mutations that impair the interaction with either or both SAM1 and SAM2 domains of Liprin-α on presynaptic differentiation. Among postsynaptic organizer proteins that induce presynaptic

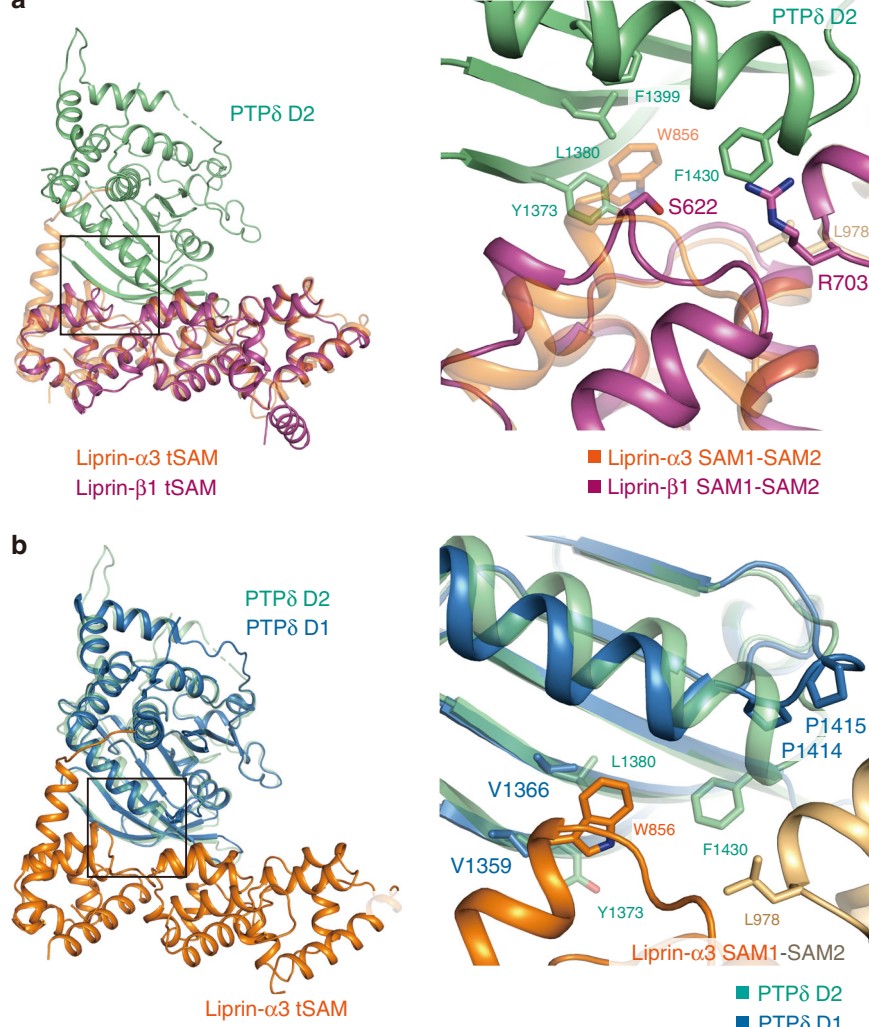

**Fig. 3 Structural comparisons between Liprin-α and -β and between PTPδ D2 and D1. a** Superposition of Liprin-β1 (purple) on Liprin-α3 (orange) bound to PTPδ D2 (green). The residues involved in the PTPδ D2–Liprin-α3 tSAM interaction and the corresponding residues of Liprin-β1 are shown as sticks in the right panel. **b** Superposition of PTPδ D1 (blue) on PTPδ D2 (green) bound to Liprin-α3 tSAM (SAM1, orange; SAM2, light brown). The residues involved in the PTPδ D2–Liprin-α3 tSAM interaction and the corresponding residues of PTPδ D1 are shown as sticks in the right panel.

differentiation through IIa RPTPs, we focused on IL1RAPL1 as an inducer because its synaptogenic activity is entirely dependent on the interaction with presynaptic PTPδ[10]. The IL1RAPL1–PTPδ interaction requires two short peptide inserts (meA-peptide and meB-peptide) within the Ig-like domains of PTPδ, which are derived from mini-exons A and B of *Ptprd* gene encoding PTPδ protein[10,18]. We then generated a mutant mouse line that lacks meB-peptide-containing PTPδ splice variants capable of binding to IL1RAPL1 by inserting a stop codon into mini-exon B (exon 9) of *Ptprd* gene. The *Ptprd* allele with the mini-exon B mutation is referred to as *Ptprd^{meB*}* in this study.

When cocultured with wild-type cortical neurons, beads conjugated with the recombinant extracellular domain of IL1RAPL1 strongly induced the accumulation of presynaptic reporter proteins, synaptophysin-mCherry and EGFP-Rab3, in contacting axons (Fig. 4a). However, as expected, cultured cortical neurons from the homozygous mutant mice (*Ptprd^{meB*/meB*}*) completely lacked the presynapse-inducing activity (Fig. 4b, c). Therefore, we performed add-back experiments, in which cortical neurons from the *Ptprd^{meB*/meB*}* mice were transfected with wild-type or mutated forms of PTPδ linked to EGFP-Rab3 by a self-cleaving P2A peptide and incubated with the

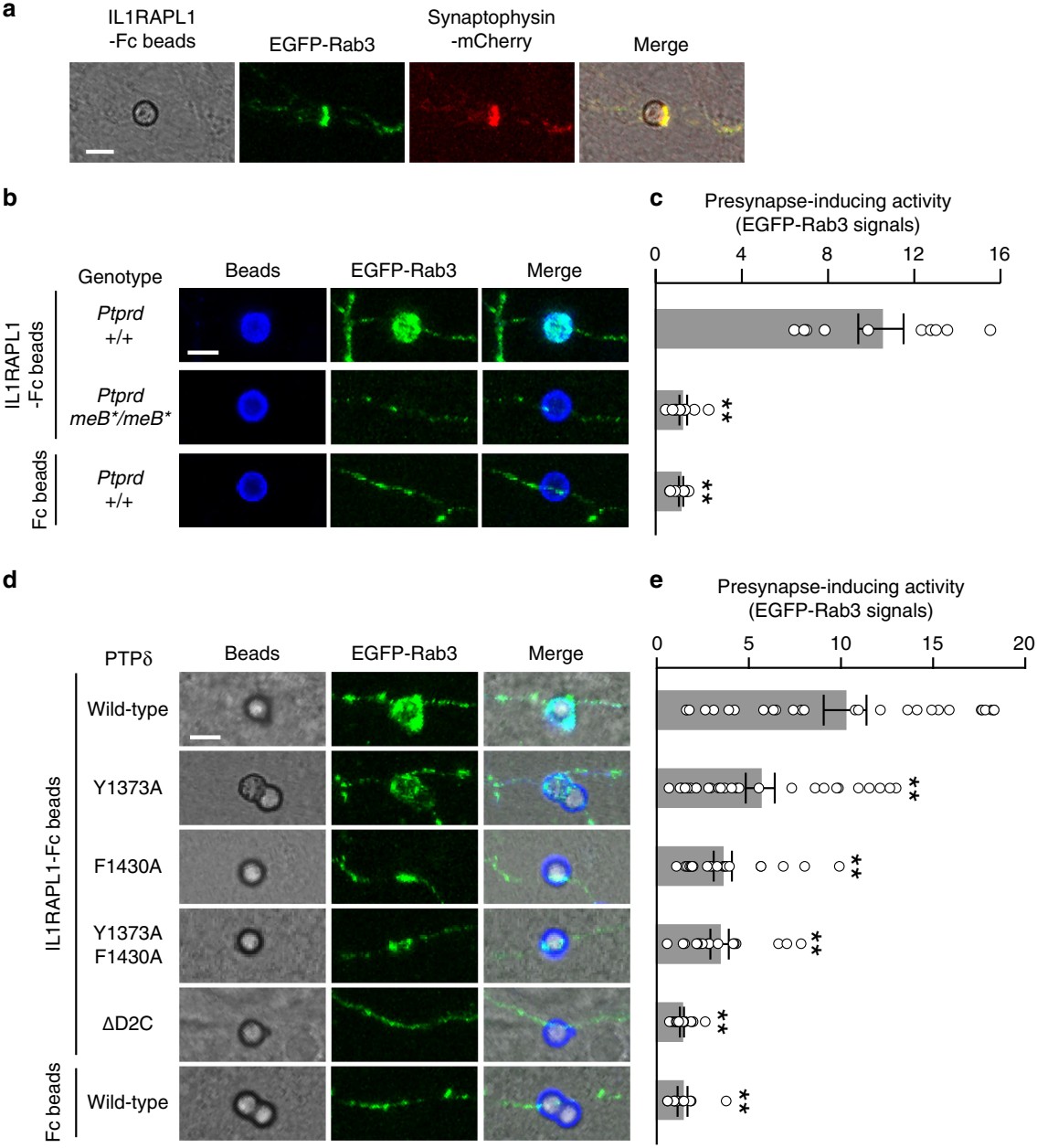

**Fig. 4 Synaptogenic activities of PTPδ mutants impairing Liprin interaction. a** IL1RAPL1-induced presynaptic differentiation of wild-type cortical neurons monitored by accumulation of EGFP-Rab3 (green) and synaptophysin-mCherry (red). **b** Complete loss of presynapse-inducing activity of IL1RAPL1 in cortical neurons from *Ptprd^{meB*/meB*}* mutant mice. **c** Relative intensity of EGFP-Rab3 signals on the surface of beads in **b** (*n* = 10 beads each). **d** Rescue of IL1RAPL1-induced EGFP-Rab3 accumulation in *Ptprd^{meB*/meB*}* cortical neurons by transient expression of wild-type and mutated forms of PTPδ. **e** Relative intensity of EGFP-Rab3 signals on the surface of beads in **d** (*n* = 27, 27, 22, 17, and 18 IL1RAPL1-Fc beads for wild type, Y1373A, F1430A, Y1373A/F1430A, and ΔD2C, respectively; *n* = 11 Fc beads for wild type). All values represent mean ± SEM. **P < 0.01 compared with wild-type neurons with IL1RAPL1 beads in **b** and compared with neurons rescued by wild-type PTPδ and cocultured with IL1RAPL1 beads in **d**; Tukey's test. Scale bars, 5 μm. Source data are provided as a Source Data file.

IL1RAPL1-conjugated beads (Fig. 4d, e). For this rescue experiments, the D2 domain mutations at the interface with Liprin-α were introduced into the PTPδ splice variant with both meA-peptide and meB-peptide insertions, which shows the highest affinity to IL1RAPL1[18]. Add-back of wild-type PTPδ restored the IL1RAPL1-induced presynaptic differentiation (*i.e.*, accumulation of EGFP-Rab3), while deletion of the D2 domain and C-terminal tail of PTPδ (ΔD2C) completely lacked the ability to rescue the defect in the presynaptic differentiation (Fig. 4d, e), suggesting that the D2 domain is required for synaptogenesis induced by PTPδ as well as by PTPσ[25,51]. The triple mutant of PTPδ for the interface with Liprin-α αN (F1503A D1504A Y1506A) restored presynaptic differentiation like wild type, in agreement with the fact that αN does not contribute to the binding to PTPδ D2 (Supplementary Fig. 6). On the other hand, the single point mutations of PTPδ at the interface with Liprin-α SAM1 (Y1373A) or SAM1/SAM2 (F1430A) disturbed the rescue activity (Fig. 4d, e), suggesting the importance of the PTPδ–Liprin-α interactions through the SAM1 and SAM2 domains for PTPδ to induce presynaptic differentiation. Tyr1373 contributes exclusively to the SAM1 interface, whereas Phe1430 does mainly to the SAM2 interface. The SAM2 interface might be more important for synaptogenic activity than the SAM1 interface. Unexpectedly, the Y1373A F1430A double mutant of PTPδ retained synaptogenic activity comparable with the F1430A mutant (Fig. 4d, e). The F1430A mutation has a larger impact than the Y1373A mutation, and may mask the impact of the Y1373A mutation in a way analogous to two factors in a linear

pathway. Otherwise, there may be another signaling pathway not mediated by Liprin-α.

## Discussion

In this study, we clarified the binding mode between PTPδ D2 and Liprin-α tSAM. Caskin is a synaptic protein, which was first identified as a CASK-binding protein from rat brain extracts. Caskin has an N-terminal ankyrin repeat domain, SH3 domain, and tSAM domain consisting of two SAM domains, SAM1 and SAM2, and is assumed to serve as an adapter molecule in synapses[52]. It has been reported that the tSAM domain of Caskin-2 binds to the D2 domain of LAR and PTPσ[23]. However, the relative configuration of SAM1 and SAM2 are obviously different between Caskin-1 and Liprin-α3. When superposing the tSAM domain of Caskin-1 (PDB 3SEI [https://doi.org/10.2210/pdb3SEI/pdb])[52] onto that of Liprin-α3 bound to PTPδ D2 using SAM1 as the reference, the position of the SAM2 domain is completely different between Caskin-1 and Liprin-α3 (Fig. 5a). Therefore, the tSAM domain of Caskin-1 binds to the D2 domain of LAR and PTPσ, likely in a manner different from the tSAM domain of Liprin-α.

The tSAM domain of Liprin-α can also bind to CASK. The insertion loop connecting the SAM1 and SAM2 domains is essential for this binding, and the isolated loop itself can bind to CASK[45]. The PTPδ D2-binding surface of Liprin-α3 tSAM is located on the opposite side of the insertion loop. When superposing the CASK–Liprin-α2 tSAM structure (PDB 3TAC [https://doi.org/10.2210/pdb3TAC/pdb]) onto the present PTPδ

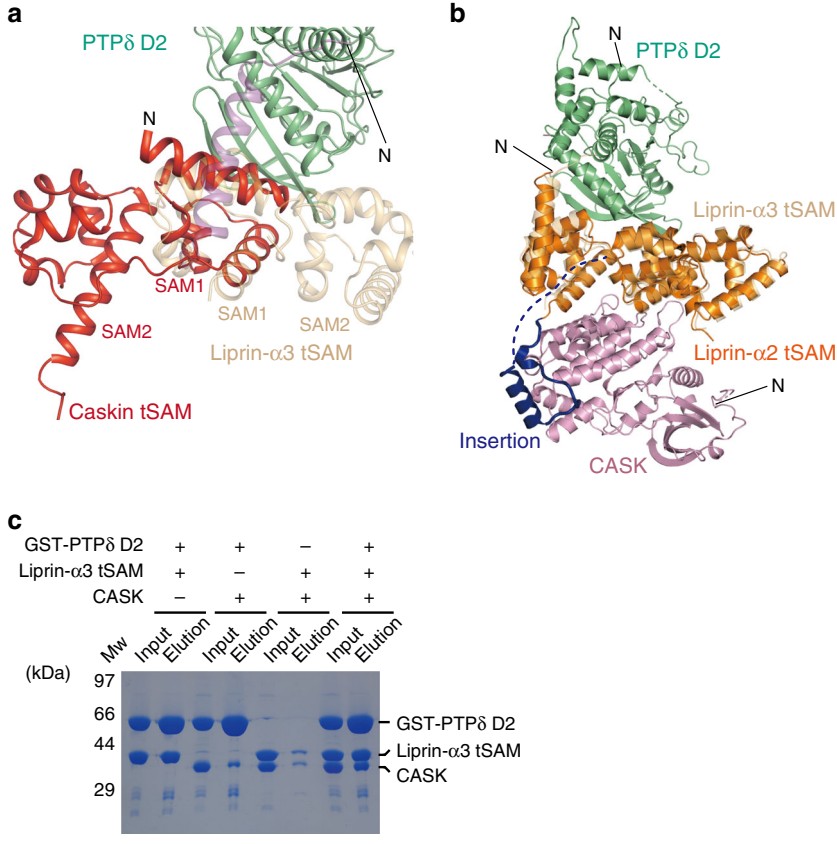

**Fig. 5 Structure-based assessment of the involvement of Caskin and CASK. a** Superposition of Caskin SAM1-SAM2 (red) on Liprin-α3 tSAM (αN, purple; SAM1-SAM2, light brown) bound to PTPδ D2 (green). **b** Superposition of the CASK (pink)–Liprin-α2 tSAM (CASK-interacting loop, dark blue; other regions, orange) complex on the PTPδ D2 (green)–Liprin-α3 tSAM (light brown) complex. No steric hindrance in this superposition suggests the formation of the tripartite signaling complex of PTPδ, Liprin-α, and CASK. **c** GST-pulldown assay for analyzing the formation of the tripartite complex suggested in **b**. Proteins bound to GST-PTPδ D2 were subjected to SDS-PAGE and stained by Coomassie Brilliant Blue. Source data are provided as a Source Data file.

D2–Liprin-α3 tSAM structure using Liprin-α tSAM as the reference, no steric hindrance occurs between CASK and PTPδ D2 (Fig. 5b). In addition, further docking of D1 on the basis of the superposition of the PTPδ D2–Liprin-α3 tSAM complex onto PTPδ D1-D2 (PDB 2FH7 [https://doi.org/10.2210/pdb2FH7/pdb]) using D2 as the reference suggests that D1 does not overlap with Liprin-α3 tSAM or CASK (Supplementary Fig. 7a). These docking analyses suggest the tripartite assembly of IIa RPTP, Liprin-α, and CASK. This idea was supported by GST-pulldown assay, where Liprin-α3 tSAM and the N-terminal kinase domain of CASK were co-precipitated with the GST-fused PTPδ D2 or D1-D2 (Fig. 5c and Supplementary Fig. 7b). The predicted geometry of the tripartite IIa RPTP–Liprin-α–CASK complex is compatible to the idea that this complex can be further assembled into other important presynaptic proteins such as RIM[37], mDia[42,43], Liprin-β[22,45], and GRIP1[53]. Such higher-order assembly of intracellular presynaptic proteins might be functionally and mechanistically linked to lateral alignments of transsynaptic connections via synaptic organizers[16,54]. Further studies on the mechanism for spatiotemporal regulation of this assembly may deepen our understanding of the molecular mechanism for synapse formation.

## Methods

**Cloning and plasmid construction.** For the structural and SPR analyses, the genes encoding mouse PTPδ D2 (residues 1213–1508) and mouse Liprin-α3 tSAM (residues 808–1114) were PCR amplified from cDNAs and cloned into the pET21a (Millipore) and pGEX-6P1 vectors (GE Healthcare), respectively. For pull-down assay, the genes encoding mouse CASK N-terminal kinase domain (residues 1–323) was cloned into the pET21a vector. PTPδ D2 (residues 1213–1508) and D1-D2 (residues 891–1508) were cloned into the pGEX-6P1 vector. For synaptogenic assay, the coding sequences of preprotrypsin signal peptide followed by FLAG tag from pFLAG-CMV-1 vector (Sigma), mouse PTPδ lacking signal peptide, porcine teschovirus-1 P2A peptide[55], EGFP, and mouse Rab3 were linked and cloned into pCAGGS vector[56] to yield pCAGGS-ppt-FLAG-PTPRD-P2A-EGFP-Rab3. The alanine substitution mutations within D2 domain were introduced by inverse PCR mutagenesis using pCAGGS-ppt-FLAG-PTPRD-P2A-EGFP-Rab3 as a template. The coding sequences of mouse synaptophysin and Rab3 were cloned into pEGFP-C1 (Clontech) and pmCherry-N1 (Clontech) vectors, respectively.

**Protein preparation.** All proteins were expressed in *Escherichia coli* Rosetta (DE3) cells (Millipore). The cells were disrupted by sonication. For the purification of PTPδ D2, the cell lysate containing PTPδ D2 fused with the C-terminal His₆-tag (PTPδ D2-His₆) was applied onto a Ni-NTA (Qiagen) column. After washing with 20 mM Tris-HCl buffer (pH 8.0) containing 300 mM NaCl and 50 mM imidazole, PTPδ D2-His₆ was eluted with 20 mM Tris-HCl buffer (pH 8.0) containing 300 mM NaCl and 250 mM imidazole. For crystallography, PTPδ D2-His₆ was further purified by size-exclusion chromatography using Superdex 200 (GE Healthcare) column with 20 mM Tris-HCl buffer (pH 7.5) containing 150 mM NaCl. For the purification of Liprin-α3 tSAM, the cells expressing Liprin-α3 tSAM fused with the N-terminal glutathione-*S*-transferase (GST-Liprin-α3 tSAM) was disrupted by sonication. The cell lysate was applied onto a Glutathione Sepharose FF column (GE Healthcare). After washing by phosphate-buffered saline (PBS), GST-Liprin-α3 was eluted with 50 mM Tris-HCl buffer (pH 8.0) containing 200 mM NaCl and 15 mM reduced glutathione. The eluted sample was further purified by anion-exchange chromatography using a Resource Q column (GE Healthcare), which was pre-equilibrated with 20 mM Tris-HCl buffer (pH 7.5) containing 50 mM NaCl. GST-Liprin-α3 was eluted with a linear gradient of 50–500 mM NaCl. The GST tag was cleaved off by HRV3c protease and removed by a Glutathione Sepharose FF column. The sample was purified again by anion-exchange chromatography in the same manner as GST-Liprin-α3. Finally, the sample was purified by size-exclusion chromatography using a Superdex 200 (GE Healthcare) column with 20 mM Tris-HCl buffer (pH 7.5) containing 150 mM NaCl.

For pull-down assay, PTPδ D2 or D1-D2 fused with an N-terminal GST tag was purified using a Glutathione Sepharose FF column, followed by size-exclusion chromatography using a Superdex 200 (GE Healthcare) column or anion-exchange chromatography using a Resource Q (GE Healthcare) column, respectively. The CASK N-terminal kinase domain fused with the N-terminal His₆-tag (His₆-CASK) was purified by Ni-affinity chromatography in the same manner as PTPδ D2-His₆. His₆-CASK was further purified by anion-exchange chromatography using a HiTrap Q HP or Resource Q column (GE Healthcare).

**Crystallization and structure determination.** For crystallization of the PTPδ D2–Liprin-α3 tSAM complex, PTPδ D2-His₆ (90 μM) and Liprin-α3 tSAM

(70 μM) were mixed. Initial crystallization screening was performed using the sitting drop vapor diffusion method at 20 °C with a Mosquito liquid-handling robot (TTP Lab Tech). About 500 conditions were tested with crystallization reagent kits supplied by Hampton Research and Qiagen. After optimization of the crystallization condition, diffraction-quality crystals of the complex were obtained with the reservoir solution containing 15% PEG 3350 and 0.1 M Tris-HCl (pH 8.5). The crystals were soaked in the reservoir solution supplemented with 30% PEG 400 and then flash-frozen in liquid N₂.

Diffraction data sets were collected at 100 K at BL41XU in SPring-8 and processed with HKL2000[57] and the CCP4 program suite[58]. The structure of the PTPδ D2–Liprin-α3 tSAM complex was determined by molecular replacement with the program Molrep[59] using PTPσ D2 (PDB 2FH7[https://doi.org/10.2210/pdb2FH7/pdb])[46] and Liprin-α2 tSAM (PDB 3TAD[https://doi.org/10.2210/pdb3TAD/pdb])[45] as the search models. The atomic model was manually improved using the program Coot[60], and refined using the program Phenix[61] with good stereochemistry (97.1% and 0.2% of residues in favored and outliers, respectively). Data collection and refinement statistics are summarized in Table 1. The buried surface area was calculated using the program PISA[62]. Sequence alignment was performed using ClustalX software[63]. All structure figures were prepared using the program PyMOL (Schrödinger, LLC; https://pymol.org/2/).

**SPR analysis.** SPR experiments were performed using Biacore T200 (GE Healthcare) at 25 °C in 10 mM Hepes-NaOH buffer (pH 7.5) containing 150 mM NaCl, and 0.005% Tween-20. Wild-type or mutant PTPδ D2-His₆ was immobilized on a CM5 sensor chip by the amine-coupling method. The amount of the immobilized ligand for each experiment is shown in response units in Table 2. The wild-type or mutant Liprin-α3 tSAM was prepared in a twofold serial dilution series from 40 μM to 39 nM. Each sample was injected in order of increasing concentration for 120 s at a flow rate of 30 μL min⁻¹, followed by a 600-s dissociation phase. The PTPδ D2-immobilized sensor chip was regenerated by 20 mM Hepes-NaOH buffer (pH 7.5) containing 1 M NaCl. All mutants examined by SPR analysis were confirmed to behave as wild type in size-exclusion chromatography (Supplementary Fig. 8). Equilibrium dissociation constants ($K_D$) were calculated using Biacore T200 software. Data are shown as mean ± standard deviation from at least three independent experiments for each sample.

**Pull-down assay.** For Fig. 5c, Liprin-α3 tSAM and/or His₆-CASK were mixed with GST-PTPδ D2 at an equimolar ratio (20 μM) in PBS with 0.1% Triton X-100 (PBS-T), and then immobilized onto Glutathione Sepharose FF (GE Healthcare) beads. The beads were washed with PBS-T three times. The bound protein complexes were then eluted with 50 mM Tris-HCl buffer (pH 8.0) containing 200 mM NaCl and 15 mM reduced glutathione. The eluted sample was separated from beads by centrifuge and subjected to SDS-PAGE analysis with Coomassie brilliant blue staining. For Supplementary Fig. 7, GST-PTPδ D2 or D1-D2 were immobilized onto Glutathione Sepharose FF (GE Healthcare) beads (10 μg of protein per 1 μL beads), and then mixed with 13 μM His₆-CASK in PBS-T with or without 29 μM Liprin-α3 tSAM. The beads were washed with PBS-T twice. The bound protein complexes were then eluted with 2xSDS loading buffer and subjected to SDS-PAGE analysis with Coomassie brilliant blue staining.

**Generation of Ptprd^meB* mice.** The electroporation of guide RNA, Cas9 protein, and ssODN into C57BL/6N mouse zygotes was performed as previously described[64]. The sequences of guide RNA (gRNA-meB) and ssODN (meB*-ssODN) were 5′-CAUCCCUCAGAGCUGCGAGA-3′ and, 5′-CCTTTTCCTCATTTCATTGTG TTCTGCATCAAACCCCCCTACATCCCTCAGAGCTCTGAGAAGGTTGGT GTGTTTTTTACTTTTTTACCCACCTTTACAAAACTACTACTT-3′, respectively. The genomic DNAs from the F0 mice were subjected to cleaved amplified polymorphic sequence analyses using primer set, 5′-ATGGTGACCTCCTTTGCTG-3′ and 5′-TCATGCATTGCATTTGGACG-3′, and restriction enzyme *Sac*I for the detection of the ssODN-mediated knock-in of the stop codon into meB. The F0 mosaic mice were crossed with wild-type C57BL/6N mice to generate heterozygous (*Ptprd^+/meB*) F1 mice. The strain was maintained by crossing *Ptprd^+/meB* females with wild-type C57BL/6N mice. The F3 heterozygous male and female mice were mated to generate homozygous mutant mice for preparation of neuronal cultures. All experimental protocols for animal studies were approved by the Animal Experiment Committee of the University of Toyama (Authorization No. A2016med-140) and conducted in accordance with the Guidelines for the Care and Use of Laboratory Animals of the University of Toyama.

**Synaptogenic assay.** Cerebral cortical neurons were prepared from homozygous (*Ptprd^meB*/meB*) mutant mice at embryonic day 18 as previously described[9]. The cultured neurons were transfected with expression vectors for wild-type or mutated forms of PTPδ linked to EGFP-Rab3 by P2A peptide at days in vitro (DIV) 8, and cocultured with Fc- or IL1RAPL1-Fc-coated magnetic beads at DIV9. The cocultures were fixed and mounted for confocal microscopy at DIV10.

**Image acquisition and quantification.** Images of cocultures were collected from at least two separate experiments using a confocal microscope (TCS SP5II, Leica Microsystems; 63× water lens, zoom setting 3.0) in a blinded manner with regard

to the expression vectors transfected. Fc- or IL1RAPL1-coated beads on the EGFP-Rab3-expressing axons were randomly imaged. Z series optical sections were projected by the brightest point method and the accumulation of EGFP-Rab3 around the beads were quantified using ImageJ software[65]. The axonal EGFP-Rab3 signal intensity of the bead-contacting area was measured as a fluorescent mean density within a circle of 7-μm diameter enclosing the beads while averaged fluorescent mean density within 7-μm-diameter circles on the same axon apart from the beads was measured as a background axonal signal. Ratios of the EGFP-Rab3 fluorescent signal intensities of the bead-contacting region and background region of the same axon were calculated for Fig. 4. Data represent mean ± SEM. Statistical significance was evaluated by one-way ANOVA followed by Tukey's post hoc test.

**Statistical analysis**. Two independent experiments were included in the statistical analysis for Fig. 4. All statistical analyses were performed using Statcel2 (OMS Publishing). No statistical method was used to determine sample size. No data were excluded. There was no randomization of mice or samples before analysis, and the mice used in this study were selected based purely on availability. For multiple comparisons, one-way ANOVA followed by Tukey's post hoc tests was used (Fig. 4c, df numerator (dfn) = 2, df denominator (dfd) = 27, F value = 76.49; Fig. 4e, dfn = 5, dfd = 116, F value = 17.95; Supplementary Fig. 6b, dfn = 2, dfd = 113, F value = 35.98).

**Reporting summary**. Further information on research design is available in the Nature Research Reporting Summary linked to this article.

## Data availability
Data supporting the findings of this manuscript are available from the corresponding authors upon reasonable request. The coordinates and structure factors of the PTPδ D2–Liprin-α3 tSAM complex have been deposited in the Protein Data Bank with the accession code 6KIP [https://doi.org/10.2210/pdb6KIP/pdb]. The source data underlying Table 2, Figs. 4a–e, and 5c, and Supplementary Figs. 6b and 7b are provided as a Source Data file. Other data are available from the corresponding authors upon reasonable request.

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

## Acknowledgements
We thank the beamline staffs of BL41XU of SPring-8 (Hyogo, Japan) for technical help during data collection. This work was supported by grants from JSPS/MEXT KAKENHI (JP 19H03162 to A.Y., JP25293057 to T.Y. and JP18H03983 to S.F.), JST PRESTO to T.Y., and JST CREST (JPMJCR12M5) to S.F.

## Author contributions
M.W. performed crystallography, SPR analysis, and pull-down experiment with assistance from A.Y., S.G.-I., Y.S. and S.F. K.K. performed protein expression experiments for crystallization. A.M. and T.S. constructed the plasmids for cell biological experiments. T.Y. and A.I. performed cell biological experiments. H.I., T.Y., M.S. and H.M. generated knock-in mutant mice. A.Y., S.F. and T.Y. wrote the paper. S.F., A.Y. and T.Y. designed the study. S.F. supervised the study.

## Competing interests
The authors declare no competing interests.
