## [Peer Review File · Nature Communications]

Reviewers' Comments:

Reviewer #1:

Remarks to the Author:

NCOMMS-19-24670, Wakita et al.

Title: Structural insights into selective interaction between type IIa receptor protein tyrosine phosphatases and Liprin- α

The Authors report the high resolution complex structure of Liprin- α and the synaptic adhesion protein receptor protein tyrosine phosphatase delta (RPTPdelta) D2 intracellular domain. The work is important for understanding the function of RPTPs as presynaptic organizers and role of Liprins in this. Data presented is sound and findings scientifically highly important, while the writing at places could be improved, in particular the introduction seem still little bit like a draft of pieces. Please also check for typos throughout. The paper could be suitable for publications in Nat. Communications with the clarifications as presented below:

1. In introduction the importance and function in vertebrates should be discussed in more detail to make a clear point of the importance of these proteins, and hence the impact of the current work. Now this remains bit unclear.

Knockdown of Liprin- α 3 is mentioned while phenotype is not severe. RPTPs are discussed earlier to some extent but remains unclear to reader if RPTPs are critical for overall synapse formation (as implicated in abstract) or just in the hemisynapse assays that depend on RPTP presence. Please elaborate and correct abstract if this is not universally critical interaction for synapse formation /maturation. Liprin also is not discussed further – e.g. work by Wong et al 2018 and some others could be reviewed to provide more background (point being it seems the background literature could be presented better). The introduction doesn't read very well overall, the flow of the text should be improved and literature covered better. How exactly this interaction is significant or is it even critical for synapse formation remains therefore slightly elusive. This must be improved for acceptance for publication.

2. Structure and interface analysis: Please provide a representative electron density figure at the complex interface in the supplement, if not in the main article.

It is mentioned a single complex is seen in the asymmetric unit and no interaction suggest higher order assembly, please provide some numbers here (interaction surface, supplementary figure of packing?) or other literature/data backing up the data that this is a 1:1 complex.

Was it checked that the purified mutants behave as the wild type and are not compromised in folding / stability, thus affecting the affinity? Can you provide some kind of data on this? (SEC purification profiles, thermofluor, native PAGE?).

It is mentioned that the alphaN contributes little to affinity – where as SAM1 and SAM2 contribute more – please also give the surface area for these interfaces, and list of all contacts in Supplement preferable per interface, it is unclear are all interactions considered in the paper?

Also, the key interactions could be listed in a Table per interface in the actual text to make it easier to follow.

I would suggest that deletion of alphaN mutation could be tested to see if it really doesn't have any significance. At least, the list as mentioned should be given and the buried surface areas. What might

be the role of alphaN if it doesn't contribute to binding? Can you comment and discuss?

It appears three residues form the binding hot spot – while double mutant of the others (e.g. R816A and L808A) might be significant – and would have been good to analyse these also. This could be discussed a bit more why this might be? E876 does not appear to be presented in Figure 2 at all, or there is a spelling mistake in Table 2 (As it actually seems, please correct).

In Figure 3. For clarity please show the overall structure alignments also.

3. Mouse/Neuronal studies: Please explain the strategy with IL1RAPL1 experiment more clearly. Also Ptpdmeb* and meA- and meB- (please check spelling here on page 10, line 199, is it meB- or meB-peptide?) relationship should be clarified. Add-back experiment remains unclear: what are the PTP mutants referred to? Define PTPdelta-A9B+ - what is this variant? Page 11, reference to figure/data is lacking completely, please add.

4. Discussion:

Do not start discussion with "Another synaptic protein..." rewrite. Start with why this is of interest.

Is it possible for CASKINs that conformation might change as the complex structure is not known and it might still interact the same way as Liprin? Perhaps discuss is it likely SAM2 of CASKIN might bind also to Liprin?

I don't understand the paragraph on phosphatase activity of D1 domain p 12, line 213 onwards: first it said the activity is dispensable but then line 234-235 it is indicated that D1 cannot not be replaced with D2? And that D1 is important "possibly through dephosphorylation... etc" Can you clarify this – seems likely two contradictory statements?

In fig 5b the "N" is not visible – change orientation or colouring so that it's actually visible from the image what is pointed at.

End of Discussion p 13, lines 215-254: it's an overstatement that further studies on this assembly would give "complete understanding of the ... synapse formation" please revise.

Other points:

R.m.s.d calculations should list how many residues out of all in the structure were aligned.

p.5 line 84 "... are a majority in the brain" Majority of what? Please check the language.

p.8 line 162 – do you really consider 36% sequence identity high? I would revise this.

p. 11 line 212 "demonstrating..." and line 220 "implying" – this is discussion, not results, please rephrase a bit and move to discussion?

Methods: the plasmid construction for cell biology experiments as is listed in the author contributions for two authors is not written at all? Please clarify the vectors used and their construction (I presume for the synaptogenic assay).

For SPR analysis (p.17) give the lowest concentration of the dilution series used.

Reviewer #2:

Remarks to the Author:

The interaction between the type IIa receptor protein tyrosine phosphatases and Liprins in presynaptic terminals is critical for synapse development and functions. In this manuscript, Wakita et al solved the high resolution structure of the pseudo-phosphatase domain (D2) of PTP δ in complex with the C-terminal three SAM domain tandem (tSAM) of Liprin- α 3. The structure reveals the mechanistic basis governing the binding between PTP δ and Liprin- α 3. The authors validated the structure using an in vitro co-culture system.

This is a straightforward and relatively simple manuscript. The results presented here are clean and solid. The results fill in an important gap in linking how presynaptic membrane spanning PTPs engage their down stream molecules in orchestrating molecular complexes critical for synaptogenesis. Therefore, it is a strong candidate for NC. I have several comments that should be within easy reach of the authors (as the authors have all of the reagents in hands), and such information will likely further improve the quality of the manuscript.

1. In the complex structure shown in Fig. 2, the N-terminal helix (α N) of liprin tSAM is in contact with PTP D2. However, mutation of residues in α N of liprin or corresponding contacting residues from PTP D2 do not seem to have obvious effect on the binding. The authors should probe further on this issue. It will be informative if the authors can perform the following additional experiments: i) assess the impact of deleting α N on the interaction between Liprin- α 3 and PTP δ both using D2 and D1-D2 tandem as the authors proposed that α N may contact D1 in the D1-D2 tandem (Fig. 5b); ii) the authors did not describe how the particular construct of tSAM was derived. Whether α N might be playing a role of linking SAM1-3 with the N-terminal helical regions of liprins; 3) although probing roles of specific residues within α N of Liprin- α 3 using the co-culture model may be problematic due to potential redundancy of liprins in neurons, the authors should look into whether mutations of F1053, Y1056 and D1054 (or their combinations) might alter presynaptic marker clustering using their co-culture system.
2. The structural comparison in Fig. 3a provides a nice indication as to why Liprin- β may not bind to PTP δ . It will be interesting to test whether substituting S622R703 of Liprin- β 1 with the corresponding residues (Trp856 and Leu978, respectively) in Liprin- α 3 might convert Liprin- β into PTP δ binder.
3. Related to Fig. 5. It will be informative and also more relevant to the real cellular setting if the author can assay formation of the GST-PTP δ -D1-D2/Liprin- α 3 tSAM/CASK complex and compare with the data with the GST-PTP δ -D2/Liprin- α 3 tSAM/CASK complex already shown in the figure.
4. Figure 4 and text line 219-220. The authors wrote "Unexpectedly, the Y1373A F1430A double mutant of PTP δ retained synaptogenic activity comparable to the F1430A mutant, implying another signaling pathway not mediated by Liprin- α ". This interpretation is not necessary true. The authors assume that the two point mutations combined should be cumulative of the individual mutations. An alternative interpretation can be that the F1430A mutation has a larger impact than the Y1373A mutation, so the latter is masked by the F1403A mutation in a way analogous to two factors in a linear pathway.

Re: NCOMMS-19-24670
Point-by-point responses

We are grateful for the reviewers' comments and made best efforts to address them. We believe that the changes based on the comments have greatly improved the manuscript.

Comments from Reviewer #1:

1. In introduction the importance and function in vertebrates should be discussed in more detail to make a clear point of the importance of these proteins, and hence the impact of the current work. Now this remains bit unclear.

Knockdown of Liprin- α 3 is mentioned while phenotype is not severe. RPTPs are discussed earlier to some extent but remains unclear to reader if RPTPs are critical for overall synapse formation (as implicated in abstract) or just in the hemisynapse assays that depend on RPTP presence. Please elaborate and correct abstract if this is not universally critical interaction for synapse formation /maturation.

Constitutive knockout of PTP δ or PTP σ has been reported to increase neonatal mortality and show severe effect on axon targeting. Their conditional knockout should be required to rigorously examine how much PTP δ or PTP σ contributes to synaptogenesis *in vivo*, but has not yet been reported. Therefore, we removed the term “major” or “representative” to mention the functional role of IIA RPTPs, and just stated “Type IIA receptor protein tyrosine phosphatases (IIA RPTPs) function as presynaptic organizers” in Abstract and the first paragraph of Introduction. We also mentioned the relationship between dysfunctions of IIA RPTPs and their postsynaptic partners and neurodevelopmental disorders such as autism to show the importance of IIA RPTPs in the neuronal system.

Liprin also is not discussed further – e.g. work by Wong et al 2018 and some others could be reviewed to provide more background (point being it seems the background literature could be presented better). The introduction doesn't read very well overall, the flow of the text should be improved and literature covered better. How exactly this interaction is significant or is it even critical for synapse formation remains therefore slightly elusive. This must be improved for acceptance for publication.

We provided more background of vertebrate Liprin in the 4th paragraph of Introduction as follows:

“... Although all four isoforms of Liprin- α are expressed in the brain with differential distribution and increase the complexity of their neuronal functions, Liprin- α 2 and - α 3 are predominantly and specifically expressed in the brain (Serra-Pages et al., J. Biol. Chem., 1998; Spangler et al., J. Comp. Neurol., 2011; Zurner et al., J. Comp. Neurol.,

2011). Knockdown of Liprin- α 2 by shRNA alters synaptic vesicle pool size and presynaptic ultrastructure. Liprin- α 2 regulates the turnover of the active zone proteins, CASK and RIM1/2, to facilitate synaptic transmission (Spangler et al., J. Cell. Biol., 2013). Superresolution microscopy revealed substantial difference in the localization of Liprin- α 2 and - α 3 inside the presynapse (Wong et al., PNAS, 2018). Liprin- α 3 is substantially colocalized with the active zone proteins, whereas Liprin- α 2 is localized to more internal region of the nerve terminals. Liprin- α 3–knockout mice generated by CRISPR/Cas9 gene editing show impaired exocytosis of synaptic vesicles, although they survive and develop mostly normally, possibly due to a complementary function of Liprin- α 2 (Wong et al., PNAS, 2018). In fact, the depletion of Liprin- α 3 causes the translocation of Liprin- α 2 to the active zone, though their functions may not be completely overlapped (Spangler et al., J. Cell. Biol., 2013; Wong et al., PNAS, 2018).

2. Structure and interface analysis: Please provide a representative electron density figure at the complex interface in the supplement, if not in the main article.

The representative electron density figure at the complex interface was shown in Supplementary Fig. 1a.

It is mentioned a single complex is seen in the asymmetric unit and no interaction suggest higher order assembly, please provide some numbers here (interaction surface, supplementary figure of packing?) or other literature/data backing up the data that this is a 1:1 complex.

The crystal packing was shown in Supplementary Fig. 1b.

Was it checked that the purified mutants behave as the wild type and are not compromised in folding / stability, thus affecting the affinity? Can you provide some kind of data on this? (SEC purification profiles, thermofluor, native PAGE?).

All mutant samples used for SPR analyses were checked by size-exclusion chromatography. The data were presented in Supplementary Fig. 8

It is mentioned that the alphaN contributes little to affinity – where as SAM1 and SAM2 contribute more – please also give the surface area for these interfaces, and list of all contacts in Supplement preferable per interface, it is unclear are all interactions considered in the paper? Also, the key interactions could be listed in a Table per interface in the actual text to make it easier to follow.

The result of the interface analysis by the program PISA was summarized as the list in Supplementary Table 2.

I would suggest that deletion of alphaN mutation could be tested to see if it really doesn't have any significance. At least, the list as mentioned should be given and the buried surface areas. What might be the role of alphaN if it doesn't contribute to binding? Can you comment and discuss?

The C-terminal half of α N hydrophobically interacts with SAM1 and the α N-deletion mutant could not be produced as a soluble form. We then prepared the N-terminally 9-residue deletion mutant (containing residues 817–1114), which should completely lack the D2-interacting region in α N, and tested its binding to PTP δ D2 by SPR analysis. Consistent with the analysis of the point mutants in α N, this deletion showed little effect on the affinity ($K_d = 17.5 \mu\text{M}$). At the moment, we have little information to consider the role of α N in the context of intermolecular interactions, and will not discuss it in this manuscript.

It appears three residues form the binding hot spot – while double mutant of the others (e.g. R816A and L808A) might be significant – and would have been good analyse these also. This could be discussed a bit more why this might be?

Synaptogenic assay to assess the Liprin- α α N–PTP δ D2 interaction showed that this interaction does not contribute to the induction of synapse formation. Therefore, we will not discuss it in this manuscript.

E876 does not appear to be presented in Figure 2 at all, or there is a spelling mistake in Table 2 (As it actually seems, please correct).

“E876A” was corrected to “E976A” in the revised Table 2.

In Figure 3. For clarity please show the overall structure alignments also.

The overall structures were added in Figure 3 accordingly.

3. Mouse/Neuronal studies: Please explain the strategy with IL1RAPL1 experiment more clearly. Also Ptp^{rdmeb} and meA- and meB- (please check spelling here on page 10, line 199, is it meB- or meB-peptide?) relationship should be clarified. Add-back experiment remains unclear: what are the PTP mutants referred to? Define PTPdelta-A9B+ - what is this variant? Page 11, reference to figure/data is lacking completely, please add.*

We revised the descriptions about the design of add-back experiments with IL1RAPL1 and neurons from PTP δ mutant mouse line, relationship between *Ptprd*^{meB*} mouse line and meA and meB, and the PTP δ splice variant used in the add-back experiments to be intelligible as follows:

page 11 lines 9-16, “Among postsynaptic organizer proteins that induce presynaptic differentiation through Ila RPTPs, we focused on IL1RAPL1 as an inducer because its synaptogenic activity is entirely dependent on the interaction with presynaptic PTP δ (Yoshida et al., 2011). The IL1RAPL1–PTP δ interaction requires two short peptide inserts (meA-peptide and meB-peptide) within the Ig-like domains of PTP δ , which are derived from mini-exons A and B of *Ptprd* gene encoding PTP δ protein (Yoshida et al., 2011; Yamagata et al., 2015). We then generated a mutant mouse line that lacks meB-peptide-containing PTP δ splice variants capable of binding to IL1RAPL1 by inserting a stop codon into mini-exon B (exon 9) of *Ptprd* gene. The *Ptprd* allele with the mini-exon B mutation is referred to as *Ptprd*^{meB*}.”

page 12 lines 2-4, “For this rescue experiments, the D2 domain mutations at the interface with Liprin- α were introduced into the PTP δ splice variant with both meA-peptide and meB-peptide insertions, which shows the highest affinity to IL1RAPL1 (Yamagata et al., 2015).

We also added references to figure in pages 11 and 12.

4. Discussion:

Do not start discussion with “Another synaptic protein...” rewrite. Start with why this is of interest.

We started Discussion with the introduction of Caskin as follows:

“Caskin is a synaptic protein, which was first identified as a CASK-binding protein from rat brain extracts. Caskin has an N-terminal ankyrin repeat domain, SH3 domain, and tSAM domain consisting of two SAM domains, SAM1 and SAM2 and is assumed to serve as an adaptor molecule in synapses (Stafford et al., Structure, 2005)”

Is it possible for CASKINs that conformation might change as the complex structure is not known and it might still interact the same way as Liprin? Perhaps discuss is it likely SAM2 of CASKIN might bind also to Liprin?

We have no idea about the conformational change or D2-interacting region of Caskins. Only what we can answer is that Caskins bind to D2 in a manner different from Liprin- α as mentioned in the last sentence of the first paragraph of Discussion.

I don't understand the paragraph on phosphatase activity of D1 domain p 12, line 213 onwards: first it said the activity is dispensable but then line 234-235 it is indicated that D1 cannot not be replaced with D2? And that D1 is important "possibly through dephosphorylation... etc" Can you clarify this – seems likely two contradictory statements?

We agree that the discussion about the role of D1 was contradictory, and therefore, removed it in the revised manuscript.

In fig 5b the "N" is not visible – change orientation or colouring so that its actually visible from the image what is pointed at.

The previous Fig. 5b was removed. A similar figure is shown in Supplementary Fig. 7a, where the N-terminal part was shown as a thicker tube.

End of Discussion p 13, lines 215-254: its an overstatement that further studies on this assembly would give "complete understanding of the ... synapse formation" please revise.

This was rephrased as follows:

“... Further studies on the mechanism for spatiotemporal regulation of this assembly may deepen our understanding of the molecular mechanism for synapse formation.”

Other points:

R.m.s.d calculations should list how many residues out of all in the structure were aligned.

The requested information was listed as Supplementary Table 1.

p.5 line 84 "... are a majority in the brain" Majority of what? Please check the language.

This phrase was removed.

p.8 line 162 – do you really consider 36% sequence identity high? I would revise this.

“High” was revised to “some”.

p. 11 line 212 “demonstrating...” and line 220 “implying” – this is discussion, not results, please rephrase a bit and move to discussion?

Interpretation of the result helps the authors to understand what the data mean. We agree that the phrases are like discussion, but would like to put them in the Results section.

In the former expression, “..., demonstrating that ...” was rephrased to “..., suggesting that ...”. In the latter expression, according to the comment from Reviewer #2, the phrase was changed as follows:

“... The F1430A mutation has a larger impact than the Y1373A mutation, and may mask the impact of the Y1373A mutation in a way analogous to two factors in a linear pathway. Otherwise, there may be another signaling pathway not mediated by Liprin- α .”

Methods: the plasmid construction for cell biology experiments as is listed in the author contributions for two authors is not written at all? Please clarify the vectors used and their construction (I presume for the synaptogenic assay).

The method of plasmid construction for the synaptogenic assay was added in the Methods section.

For SPR analysis (p.17) give the lowest concentration of the dilution series used.

The lowest concentration (39 nM) was indicated in the Methods section.

Comments from Reviewer #2:

1. In the complex structure shown in Fig. 2, the N-terminal helix (α N) of liprin tSAM is in contact with PTP D2. However, mutation of residues in α N of liprin or corresponding contacting residues from PTP D2 do not seem to have obvious effect on the binding. The authors should probe further on this issue. It will be informative if the authors can perform the following additional experiments:

i) assess the impact of deleting α N on the interaction between Liprin- α 3 and PTP δ both using D2 and D1-D2 tandem as the authors proposed that α N may contact D1 in the

D1-D2 tandem (Fig. 5b);

The C-terminal half of α N hydrophobically interacts with SAM1 and the α N-deletion mutant could not be produced as a soluble form. We then prepared the N-terminally 9-residue deletion mutant (containing residues 817–1114), which should completely lack the D2-interacting region in α N, and tested its binding to PTP δ D2 and D1-D2 by SPR analysis. Consistent with the analysis of the point mutants in α N, this deletion showed little effect on the affinity ($K_d = 17.5 \mu\text{M}$ and $13.3 \mu\text{M}$, respectively).

We decided to remove the discussion about the role of D1 because it is hard to discuss it without contradiction. Therefore, we will only show the binding data for D2 in the revised manuscript.

ii) the authors did not describe how the particular construct of tSAM was derived. Whether α N might be playing a role of linking SAM1-3 with the N-terminal helical regions of liprins;

We designed the expression constructs of Liprin- α tSAMs, based on the structures of Liprin- α 2 tSAM (Wei et al., Mol. Cell, 2011). This was mentioned in the middle of the first paragraph of the subsection “Overall structure of the PTP δ D2–Liprin- α 3 tSAM complex”.

We also suppose that α N might be playing a role of linking SAM1-3 with the N-terminal helical regions of liprins, but have had no experimental evidence so far.

3) although probing roles of specific residues within α N of Liprin- α 3 using the co-culture model may be problematic due to potential redundancy of liprins in neurons, the authors should look into whether mutations of F1053, Y1056 and D1054 (or their combinations) might alter presynaptic marker clustering using their co-culture system.

We tested the synaptogenic activity of the triple mutant F1503A D1504A Y1506A, and confirmed that it has the activity comparable to wild type (Supplementary Fig. 6).

2. The structural comparison in Fig. 3a provides a nice indication as to why Liprin- β may not bind to PTP δ . It will be interesting to test whether substituting S622R703 of Liprin- β with the corresponding residues (Trp856 and Leu978, respectively) in Liprin- α 3 might convert Liprin- β into PTP δ binder.

We tested whether Liprin- β 1 tSAM (S622W R703L) was able to bind to PTP δ D2 by pulldown analysis but failed to detect the binding (see below). This analysis was repeated twice independently.

3. Related to Fig. 5. It will be informative and also more relevant to the real cellular setting if the author can assay formation of the GST-PTP δ -D1-D2/Liprin- α 3 tSAM/CASK complex and compare with the data with the GST-PTP δ -D2/Liprin- α 3 tSAM/CASK complex already shown in the figure.

We confirmed the tripartite assembly of GST-PTP δ -D1-D2, Liprin- α 3 tSAM, and CASK by pull-down assay (Supplementary Fig. 7b). The docking model of PTP δ -D1-D2, Liprin- α 3 tSAM, and CASK using the structures of the CASK–Liprin- α 2 tSAM complex and PTP σ D1-D2 were also shown in Supplementary Fig. 7a.

4. Figure 4 and text line 219-220. The authors wrote “Unexpectedly, the Y1373A F1430A double mutant of PTP δ retained synaptogenic activity comparable to the F1430A mutant, implying another signaling pathway not mediated by Liprin- α ”. This interpretation is not necessary true. The authors assume that the two point mutations combined should be cumulative of the individual mutations. An alternative interpretation can be that the F1430A mutation has a larger impact than the Y1373A mutation, so the latter is masked by the F1403A mutation in a way analogous to two factors in a linear pathway.

The suggested phrase was revised as follows:

“... Unexpectedly, the Y1373A F1430A double mutant of PTP δ retained synaptogenic activity comparable to the F1430A mutant. The F1430A mutation has a larger impact than the Y1373A mutation, and may mask the impact of the Y1373A mutation in a way analogous to two factors in a linear pathway. Otherwise, there may be another signaling pathway not mediated by Liprin- α .”

Reviewers' Comments:

Reviewer #1:

Remarks to the Author:

Thank you for the authors for the corrections. I see the paper is now improved and would recommend publication.

Two notes, in figure 1: fig. 1c appears smaller than Fig 1b and the pseudo-catalytic site is poorly visible, please improve this figure for the reader - should be same size as Fig. 1b and residues clearly visible (might add a zoom in if needed).

Also, as one note for Supplementary figure 1 - Crystal packing should be shown along a crystallographic axis with the unit cell or symmetry axis shown in the figure, now the orientation seems random and thus is not informative.

Other than that the authors have answered all the questions in satisfactory way.

Reviewer #2:

Remarks to the Author:

The authors have adequately addressed my comments both with experiments and with text revisions. The revised manuscript is with improved quality. I support the publication of the paper in NC.

Re: NCOMMS-19-24670A

Comments from Reviewer #1

Two notes, in figure 1: fig. 1c appears smaller than Fig 1b and the pseudo-catalytic site is poorly visible, please improve this figure for the reader - should be same size as Fig. 1b and residues clearly visible (might add a zoom in if needed).

The size of Fig. 1c was increased to a similar size of Fig. 1b. In addition, the font size of the labels was decreased, and the labels were placed so as to make the pseudo-catalytic site clearly visible.

Also, as one note for Supplementary figure 1 - Crystal packing should be shown along a crystallographic axis with the unit cell or symmetry axis shown in the figure, now the orientation seems random and thus is not informative.

The unit cell with the crystallographic axis and origin was added to Supplementary Fig. 1d. The orientation was changed so as to make the crystallographic axis and origin visible.